# Temperature-Sensitive Auditory Neuropathy: Report of a Novel Variant of OTOF Gene and Review of Current Literature

**DOI:** 10.3390/medicina59020352

**Published:** 2023-02-13

**Authors:** Francesca Forli, Silvia Capobianco, Stefano Berrettini, Luca Bruschini, Silvia Romano, Antonella Fogli, Veronica Bertini, Francesco Lazzerini

**Affiliations:** 1ENT, Audiology and Phoniatrics Unit, University of Pisa, 56124 Pisa, Italy; 2Division of ENT Diseases, Karolinska Institutet, 171 77 Stockholm, Sweden; 3Department of Medical and Oncological Area, Section of Medical Genetics, Azienda Ospedaliero Universitaria Pisana, 56124 Pisa, Italy; 4Department of Laboratory Medicine, Section of Molecular Genetics, Azienda Ospedaliero Universitaria Pisana, 56124 Pisa, Italy; 5Department of Laboratory Medicine, Section of Cytogenetics, Azienda Ospedaliero Universitaria Pisana, 56124 Pisa, Italy

**Keywords:** temperature-sensitive auditory neuropathy, OTOF, otoferlin, deafness, targeted next-generation sequencing

## Abstract

*Background and objectives*: Otoferlin is a multi-C2 domain protein implicated in neurotransmitter-containing vesicle release and replenishment of the cochlear inner hair cell (IHC) synapses. Mutations in the OTOF gene have been associated with two different clinical phenotypes: a prelingual severe-to-profound sensorineural hearing loss (ANSD-DFNB9); and the peculiar temperature-sensitive auditory neuropathy (TS-ANSD), characterized by a baseline mild-to-moderate hearing threshold that worsens to severe-to-profound when the body temperature rises that returns to a baseline a few hours after the temperature has fallen again. The latter clinical phenotype has been described only with a few OTOF variants with an autosomal recessive biallelic pattern of inheritance. Case report: A 7-year-old boy presented a picture compatible with TS-ANSD exacerbated by febrile states or physical exercise with mild-to-moderate hearing loss at low and medium frequencies and a decrease in speech discrimination that worsened with an unfavorable speech-to-noise ratio. Otoacoustic emissions (OAEs) were present whereas auditory brainstem responses (ABRs) evoked by a click or tone-burst were generally absent. No inner ear malformations were described from the CT scan or MRI. Next-generation sequencing (NGS) of the known deafness genes and multi-phasic bioinformatic analyses of the data detected in OTOF a c.2521G>A missense variant and the deletion of 7.4 Kb, which was confirmed by array-comparative genomic hybridization (array-CGH). The proband’s parents, who were asymptomatic, were tested by Sanger sequencing and the father presented the c.2521G>A missense variant. *Conclusions*: The picture presented by the patient was compatible with OTOF-induced TS-ANSD. OTOF has been generally associated with an autosomal recessive biallelic pattern of inheritance; in this clinical report, two pathogenic variants never previously associated with TS-ANSD were described.

## 1. Introduction

Temperature-sensitive hearing loss (TSHL) has been puzzling researchers for years. It still represents a unique phenotype in audiology and is characterized by a worsening of the hearing threshold as soon as the body temperature rises [1]. The only gene that is currently associated with TSHL is OTOF, located on chromosome 2. Coding for otoferlin, a multi-C2 domain protein belonging to the family of ferlins, is generally implicated in membrane–membrane fusion events [2]. Otoferlin was identified by immunofluorescence in cochlear inner hair cell (IHC) synapses, which are characterized by the peculiar synaptic ribbon anchoring a readily releasable pool of neurotransmitter-containing vesicles to allow their timely and unprecedented rate of release and replenishment [3]. The most accredited hypotheses regarding otoferlin functions involve its role as a calcium sensor for synaptic vesicle fusion and as priming factor to enable fast vesicle replenishment [2].

A normal auditory function and speech discrimination depend on the faithful decoding of information from the cochlea and transference through the auditory nerves. To achieve speech discrimination, OTOF-dependent IHC synaptic vesicle exocytosis needs to be indefatigable, highly efficient, and accurately synchronized to determine reliable and temporally precise cochlear potentials with fast rise times, shorts onsets, and short peak latencies [4,5]. Consequently, mutations of otoferlin have been associated with clinical pictures of auditory synaptopathy, with a severe impact on speech discrimination. 

Since its first description in 1999 [6], more than 220 OTOF pathogenic variants have been described [7], mostly with an autosomal recessive pattern of inheritance. In rare cases, only one parent of a proband has been proven to be a carrier for a mutated OTOF gene. The proband presents the disease either as the result of an additional de novo mutation or as the consequence of uniparental isodisomy [8]. The distribution of OTOF variants has been reported to differ among ethnic populations and geographical regions [9,10,11,12]. 

Otoferlin’s molecular structure is characterized by 6 C2 domains (C2A to F), implicated in calcium binding, and by a transmembrane domain (TMD). Calcium-binding C2 domains play a crucial role in otoferlin activity as they carry most pathogenic OTOF mutations [10]. 

OTOF-related deafness comprises two different phenotypes: OTOF-related auditory neuropathy spectrum disorder (ANSD) and temperature-sensitive auditory neuropathy spectrum disorder (TS-ANSD). ANSD involves a congenital/prelingual severe-to-profound bilateral deafness (DFNB9) without any associated malformations of the inner ear from MRI or CT scans. On the other side, TS-ANSD is characterized by normal-to-moderate hearing loss at a baseline body temperature, which turns to severe-to-profound as soon as the body temperature rises by 0.5 °C or more (as with fever or physical exercise), with a resolution within hours from the return to the baseline [13]. In both cases, otoacoustic emissions (OAEs) are generally detected at birth whereas auditory brainstem responses (ABRs) are absent or severely altered, which mark the deficit as a neuropathy/synaptopathy of IHC involving defective synaptic transmissions in the cochlea [14]. Both forms have been clinically characterized by defects in speech discrimination exacerbated by background noise. This also is a marker for a picture compatible with neuropathy or synaptopathy [14,15,16,17].

In this paper, we report on a 7-year-old boy affected by TS-ANSD, presenting two heterozygous variants in OTOF that have never been associated with temperature sensitivity. The audiological and genetic features are described in detail, and a review of the current literature is reported.

## 2. Materials and Methods

### 2.1. Clinical History

A 7-year-old boy underwent an audiological evaluation for reported hearing difficulties at the Otolaryngology, Audiology, and Phoniatrics department of Pisa University Hospital (Pisa, Italy). The patient was born full-term with presenting neonatal distress; however, this did not require intensive neonatal care. The newborn auditory screening performed with OAEs was bilaterally normal. A family history of deafness was dubious (the aunt of the maternal grandmother had presented with hearing loss at a young age). At 2 years of age, the patient experienced the first episode of transient deafness. Subsequently, numerous other episodes of transient hearing loss after physical efforts or fever were reported by the parents without additional vestibular symptoms. At 7 years of age, the boy’s language development was within a physiological range, despite a few phonetical mistakes. The patient reported hearing difficulties, especially whilst watching television or listening in noisy environments. The parents did not report any learning difficulty at school. The patient underwent an audiological evaluation by otoscopy, pure-tone and speech audiometry, tympanometry, OAEs, ABRs, a speech perception test in quiet and with background noise, and competitive adaptive testing (the Italian Simplified Matrix Test) [18]. In a condition of fever (38.7 °C) that occurred at 8 years of age, it was possible to perform pure-tone audiometry, with evidence of a progression of the sensorineural hearing loss to moderate-to-severe. Once the body temperature returned to a baseline following the administration of fever reducers (paracetamol), it was followed by a hearing threshold a few hours later. The patient was subsequently evaluated at 9 and 10 years of age and did not show a progression of the hearing threshold.

### 2.2. Audiological Evaluation

At 7 years of age, the patient underwent a complete audiological examination. Otoscopy was bilaterally normal. Pure-tone audiometry revealed mild-to-moderate sensorineural hearing loss at middle and low frequencies (500–1000 Hz) (Figure 1). Tympanometry was bilaterally normal (type A). Stapedial reflexes (SRs) were bilaterally absent at maximum stimulation levels. OAEs were successfully recorded; click-evoked and tone-burst-evoked ABRs on the left ear could not record any response whereas on the right side, an evident V-wave could not be recorded, even with a 95 dB SPL click stimulus.

Vocal audiometry detected a slight decrease bilaterally in vocal discrimination. The child’s disyllabic word recognition was 100% in a quiet environment, 90% with a signal-to-noise ratio (SNR) of + 10 and + 5, and 70% with a SNR of 0. With the Italian Simplified Matrix Test [18], the patient’s speech reception threshold (SRT) was + 5.5. The child also presented specific phonological difficulties in the discrimination of the vowels /a/ and /o/. 

At the age of 8 years, the patient had an episode of fever with a rise in body temperature to 38.7 °C. He reported a worsening of his hearing loss, which was tested by pure-tone audiometry (Figure 2), with evidence of a progression of the hearing threshold to moderate-to-severe. His disyllabic word recognition score also worsened to 60% in a quiet environment, 40% with a signal-to-noise ratio (SNR) of + 10, 30% with a SNR of + 5, and 10% with a SNR of 0. Tympanometry was normal (type A) and stapedial reflexes were again bilaterally absent. Once the body temperature returned to a baseline a few hours later, the hearing loss was also restored to the baseline levels. The audiological picture was compatible with temperature-sensitive auditory neuropathy/synaptopathy, with a mild-to-moderate sensorineural hearing loss at the baseline temperature.

### 2.3. Imaging

The patient underwent a CT scan of the inner ear without contrast together with an MRI of the temporal bone with and without contrast. Both imaging studies did not show any bilateral anatomical abnormality of the inner ear and acoustic nerves.

### 2.4. Genetic Testing

#### 2.4.1. NGS Analysis

DNA samples from the patient and his parents were collected from peripheral blood and extracted using a QIA symphony automatic instrument (QIAGEN, Germany). For the next-generation sequencing (NGS) analysis, we developed a panel using a custom-designed Agilent SureSelect QXT target sample library preparation and capture method (Agilent Technologies, Santa Clara, CA, USA). We performed a parallel high-throughput sequencing analysis of the coding exons and flanking regions of 37 genes associated with deafness: COL11A2 (NM_080680), GJB2 (NM_004004), GJB6 (NM_001110219), MYO6 (NM_004999), MYO7A (NM_000260), TECTA (NM_005422), TMC1 (NM_138691), BSND (NM_057176), CDH23 (NM_052836), ESRRB (NM_004452), GIPC3 (NM_133261), ILDR1 (NM_175924), LOXHD1 (NM_144612), MYO15A (NM_016239), OTOF (NM_001287489), PCDH15 (NM_033056), PJVK (NM_001042702), SLC26A4 (NM_000441), TMPRSS3 (NM_001256317), TRIOBP (NM_001039141), USH1C (NM_153676), WHRN (NM_015404), ACTG1 (NM_001614), COCH (NM_004086), COL11A1 (NM_001854), EYA4 (NM_004100), GJB3 (NM_024009), KCNQ4 (NM_004700), MYH9 (NM_002473), SIX1 (NM_005982), WFS1 (NM_006005), POU3F4 (NM_000307), PRPS1 (NM_001204402), FOXI1 (NM_012188), KCNJ10 (NM_002241), EYA1 (NM_000503), and SIX5 (NM_175875).

A total of 10 ng of the DNA/sample was used for the target enrichment step. The captured libraries were sequenced with a Next550 Sequencer (Illumina, San Diego, CA, USA).

#### 2.4.2. Bioinformatics Analysis

The read alignment, variant calling, and annotation were performed with Agilent SureCall software (Agilent Technologies, Santa Clara, CA, USA) and the CLC Genomics Workbench application (QIAGEN, Venlo, Netherlands), whose workflows enable the sensitive detection of single-nucleotide variants (SNVs) and insertions or deletions (Copy Number Variants, CNVs).

The sequencing coverage of each exon was analyzed in detail using the IGV tool (Integrative Genomics Viewer, Broad Institute and the Regents of the University of California, USA) to reduce the risk of incorrect results and for the detection of deletions.

The classification of the sequence variants was automatically produced in VarSome [19], which was based on an accurate analysis of NGS data from different databases, including UniProt Variants, dbscSNV, DANN, SNVs, 1000 Genomes, gnomAD, HGMD, and ClinVar. According to the American College of Medical Genetics and Genomics (ACMG) guidelines [20], the variants of interest were classified as pathogenic, likely pathogenic, benign, likely benign, or of uncertain significance. To confirm the pathogenicity of the variants of interest, we also referred to the Deafness Variation Database [21].

The suspected candidate variants were confirmed by Sanger sequencing (the primer sequences and PCR conditions are available upon request).

#### 2.4.3. Array-Comparative Genomic Hybridization (Array-CGH)

A total of 200 ng of genomic DNA was isolated from peripheral blood by standard methods. DNA from healthy subjects was used as the control (Agilent Technologies, Santa Clara, CA, USA). The test and reference DNA were differentially labeled with Cy5-dCTP or with Cy3-dCTP (cyanine 5/3-labelled 5′-triphosphate-2′-deoxycytidine) using random primer labeling and applied to 60 K arrays, according to the manufacturer’s protocol (Agilent, Santa Clara, CA, USA). The slides were washed and scanned using an Agilent scanner. The identification of individual spots on the scanned arrays and quality slide evaluation were performed with Agilent-dedicated software (Feature Extraction, Agilent, Santa Clara, CA, USA). The copy number variants (CNVs) were identified with Cytogenomics 3.0.6.6 (Agilent, Santa Clara, CA, USA) using an ADM-2 (aberration detection method-2) algorithm. The CNV analysis was performed according to the guidelines of the Italian Society of Human Genetics [22] and the American College of Medical Genetics [23].

### 2.5. Genetic Results

The targeted NGS and bioinformatic data analysis identified the following heterozygous variants in the patient in the OTOF gene (RefSeq: NM_194248.2) (Figure 3): c.2521G>A, p.(Glu841Lys);c.(897+1_898−1)_(1579+1_1580−1)del.

The c.2521G>A missense variant was validated using PCR-Sanger sequencing technology whereas the presence of the deletion was confirmed by array-CGH, which did not reveal any other clinically relevant genomic imbalances in the genome. The inheritance of these variants was then checked in the unaffected parents. The missense variant was inherited from the father. The presence of the deletion was not evaluated because the samples of the parents were not available for an array-CGH analysis.

The c.2521G>A, p.Glu841Lys variant, which occurred in exon 21, resulted in a glutamic acid-to-lysine substitution at position 841. It has previously been reported as a pathogenic allele [24,25] in association with auditory neuropathy spectrum disorder and it is classified as pathogenic in HGMD (Human Gene Mutation Database) (CM1513835) and the Deafness Variation Database, according to standard ACMG guidelines. This variant has a low allele frequency in the GnomAD database (0.0000066, rs772729658) and is well-conserved in several species; it is located between the C2C and C2D domains.

After the initial finding of only one variant in the OTOF gene for the proband, we checked the exon coverage using the IGV tool and identified that exons 10 to 14 were notably poorly covered, suggesting a heterozygous 7.4 Kb genomic deletion from intron 9 to intron 14 (chr2:26705247~26712680). This deletion was confirmed by array-CGH. This genomic deletion alters the reading frame and is supposed to lead to a putative truncated protein lacking five out of six functional C2 domains.

We found two additional heterozygous variants in the proband:c.−22−2A>C in intron 1 of the GJB2 (Gap Junction Protein Beta 2/Connexin 26) gene (RefSeq: NM_004004) (rs201895089), classified as a likely pathogenic variant;c.2717A>G, p.(Tyr906Cys) in exon 9 of the TECTA gene (Alpha-Tectorin) (RefSeq: NM_005422), classified as a variant with an uncertain significance.

The father proved to be an asymptomatic carrier of the GJB2 and TECTA variants in a heterozygous state; the mother did not carry any of them.

### 2.6. Therapeutic Approach

Considering the audiological picture of mild sensorineural hearing loss at middle and low frequencies with a decrease in speech discrimination exacerbated by background noise and worsened by an unfavorable noise-to-speech ratio, the patient was advised to wear hearing aids and to use a wireless system in the school environment. The patient and his parents reported an improvement in speech discrimination and in general hearing performance with the use of hearing aids and with the wireless system in the school environment. Considering the temperature sensitivity of the hearing threshold, the patient and his parents were advised to avoid a rise in body temperature by promptly treating fever states with fever reducers and by avoiding intense physical exercise.

## 3. Discussion

The first report of TS-ANSD came from Gorga and colleagues in 1995 [26], who described a patient presenting recurrent, but completely reversible, hearing loss during a fever. Poor speech recognition emerged earlier than pure-tone hearing loss and resolved more slowly. OAEs and electrocochleography were normal, both in afebrile and febrile states, proving a good cochlear function. However, acoustic reflexes, ABRs, and mechanically-evoked trigeminofacial reflexes were abnormal as a consequence of an alteration to the central connection with the brainstem. A second report came from Starr and colleagues three years later [1], describing a clinical picture characterized at a baseline body temperature by mild low-frequency HL with excellent speech recognition in silence and poor speech recognition in noise. Even at the baseline, the ABRs were abnormal, with a delay to waves IV and V and the absence of waves I and III. With a rise in the body temperature, pure-tone audiometry recorded severe HL with no speech recognition and absent ABRs. On the contrary, OAEs were present, both in afebrile and febrile states. In 2006, the patient described by Starr in 1998 was reported to be a carrier of an OTOF variant on one of the two alleles [27]. In a further study, it was determined that the second allele also carried a pathogenic variant [28]. 

Similar clinical reports were published in the subsequent years [9,13,27,29,30,31,32,33,34,35,36,37]. Even though several genes have been associated with auditory neuropathy, only a few OTOF variants have been associated with clinical pictures compatible with TS-ANSD [10] (Table 1). 

In 2016, Strenzke and colleagues [28] investigated the molecular basis of heat sensitivity by studying a mouse model carrying the most studied OTOF mutation. This resulted in a clinical picture that worsened as soon as the body temperature rose (p.Ile515Thr/p.Ile515Thr). However, it is worth mentioning that even wild-type otoferlin has been described as thermally unstable [28]; therefore, it is likely that a few mutations can enhance this heat sensitivity. For example, Ile515Thr involves the hydrophobic core of the C2C otoferlin domain with a substitution of appropriately hydrophobic isoleucine with the hydrophilic threonine, increasing the protein’s instability and, consequently, its ability to refold and its rate of degradation with an increase in temperature. A similar phenotype to p.Ile515Thr has been described for other mutations associated with temperature-sensitive hearing loss as all these amino acid substitutions seem to cause only slight destabilizations of one C2 domain, decreasing the chances of otoferlin refolding after heat exposure [38]. Considering the OTOF mutations associated with TS-ANSD (Figure 4), most variants seem to affect two protein areas: the domains C2C and C2D and the sequence between them (positions 515–1116), together with domains C2E and C2F (1607–1804).

This study reported the clinical features of a patient presenting with a picture compatible with TS-ANSD. 

The genetic analysis highlighted heterozygous variants in the GJB2 and TECTA genes.

The GJB2 and TECTA genes cause recessive non-syndromic deafness, and their variations have never been reported in TS-ANSD. 

Our patient also presented with two heterozygous variants in OTOF: a missense variant (c.2521G>A) in exon 21, classified as pathogenic; and a loss-of-function deletion of 7.4 Kb. 

Considering the clinical picture of mild sensorineural hearing loss at middle and low frequencies with a decrease in speech discrimination and a history of the worsening of the hearing threshold after febrile episodes or physical exercise, it is likely that the proband’s phenotype was associated with the above-described mutations as clinical pictures of TS-ANSD have been associated solely with the variants of otoferlin. The missense variant (c.2521G>A) has been reported in the literature in three subjects presenting with OTOF compound heterozygosity (one presented with p.Glu841Lys/p.Leu1011Pro; the other two presented with p.Glu841Lys/p.Arg1939Gln) [24,25]. However, these reports described a phenotype of ANSD-DFNB9 with severe-to-profound prelingual bilateral hearing loss. This variant has not yet been associated with a phenotype of TS-ANSD. 

p.Glu841 is well-conserved in several species and the p.Glu841Lys variant is consistently identified as pathogenic by in silico analyses. Its finding in ANSD patients [24,25] and in our TS-ANSD case induced us to suppose that other factors could influence the phenotypic outcome. We speculated that a 7.4 Kb loss-of-function deletion or a different genetic/environmental background could contribute to the different phenotype.

So far, only three intragenic deletions have been reported, ranging from 4 to 61.6 Kb; all of them have been associated with ANSD-DFNB9 [24,39,40]. The 7.4 Kb deletion described here skipped from exon 9 to 14 and altered the reading frame; therefore, we supposed that it would to lead to a putative truncated protein lacking five out of six functional C2 domains. This was the first deletion associated with TS-ANSD.

## 4. Conclusions

Our patient, a 7-year-old boy with a clinical picture of sensorineural mild-to-moderate bilateral hearing loss at middle–low frequencies with difficulties in speech discrimination exacerbated by a rise in body temperature (as with fever or physical exercise), was found to carry two heterozygous variants in OTOF, which have never been associated with temperature sensitivity. The clinical picture was, therefore, compatible with an OTOF-induced TS-ANSD. This is the first report of a deletion associated with TS-ANSD. For the complete characterization of complex audiological cases such as the one reported in this study, we highlight the importance of adding a genetic characterization to extensive audiological and radiological evaluations.

## Figures and Tables

**Figure 1 medicina-59-00352-f001:**
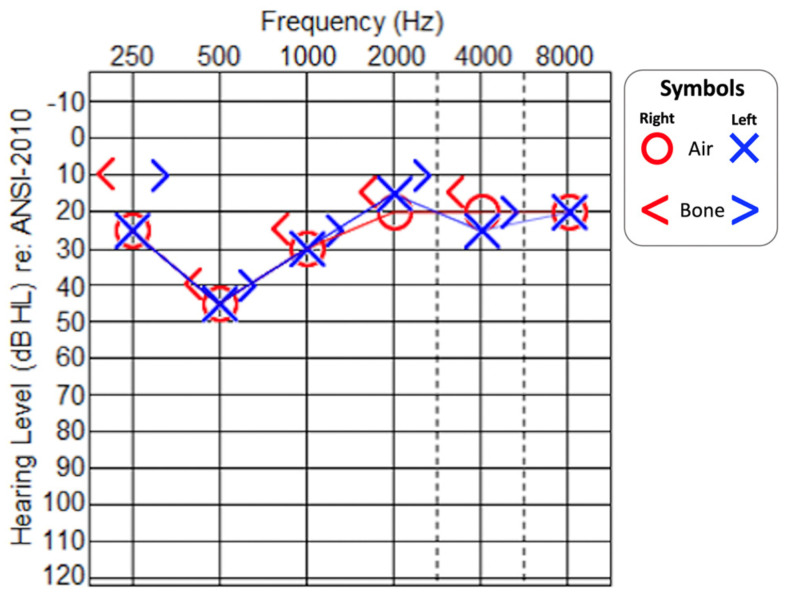
Pure−tone audiogram at 7 years of age in afebrile state. Red line: right ear. Blue line: left ear.

**Figure 2 medicina-59-00352-f002:**
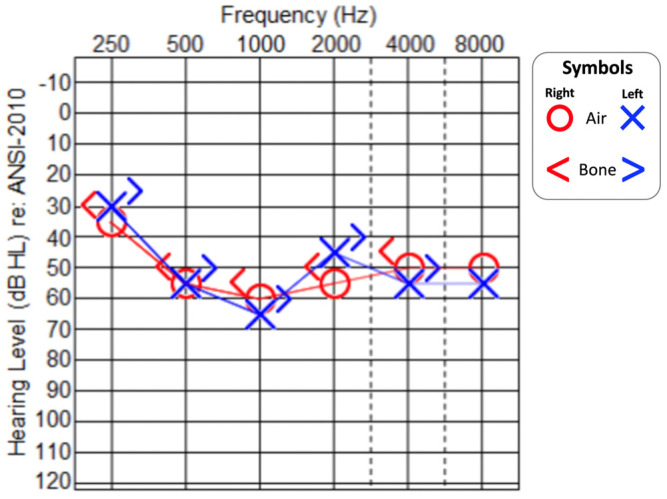
Pure−tone audiogram at 8 years of age during an episode of fever (T = 38.7 °C). Red line: right ear. Blue line: left ear.

**Figure 3 medicina-59-00352-f003:**
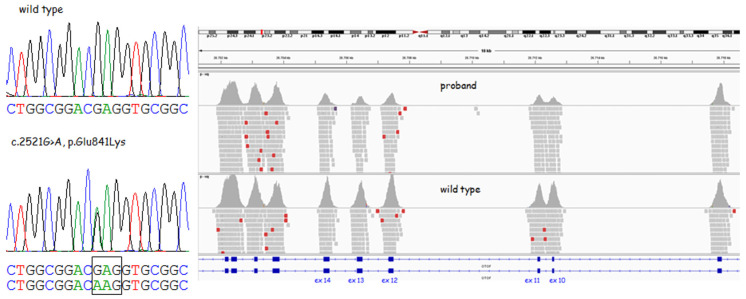
Variants in the OTOF gene in the proband. On the left are the electropherograms showing the wild-type sequence and the heterozygous missense variant (c.2521G>A). On the right are the IGV alignments showing a halved coverage from exon 10 to 14, indicating a genomic deletion (GRCh37/hg19; RefSeq: NM_194248.2). Reads that are colored red have larger than expected inferred sizes, and therefore indicate possible deletions. However, red reads are randomly present also in the wild-type control, so taken *per se* they have no clinical significance. Blue rectangles represent exons, connected by blue lines which represent introns.

**Figure 4 medicina-59-00352-f004:**
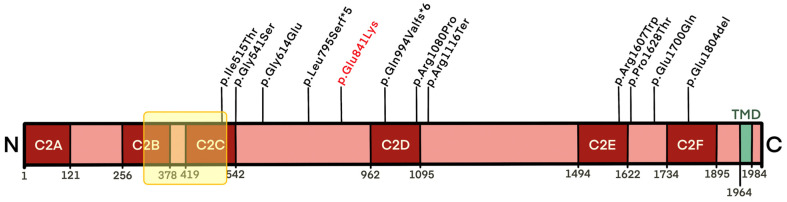
Mutations in the OTOF gene carried by TS-ANSD patients. The numbers were based on the amino acid sequence of human otoferlin and C2 domains (C2A to F) and the transmembrane domain (TMD) is depicted according to an in silico analysis [2]. The point mutation reported in the present study is indicated in red. The protein area affected by the deletion is indicated in yellow.

**Table 1 medicina-59-00352-t001:** Overview of genotype–phenotype correlations of OTOF variants in patients affected by TS-ANSD in the present study and in previous studies. PTA, pure-tone audiometry; SDS, speech discrimination score; SR, stapedial reflex; OAEs, otoacoustic emissions; ABR, auditory brainstem response; Ty, tympanometry; DWRS, disyllabic word recognition score; † before hearing improvement.

	Reference	Starr, 1998 [1] Varga, 2006 [27] Strenzke, 2016 [28]	Romanos, 2009 [31]	Marlin, 2010 [32]	Matsunaga, 2012 [9]; Kaga, 2016 [35]	Wang, 2010 [36]; Zhang, 2016 [13]	Zhang, 2016 [13]	Zhang, 2016 [13]	Zhu, 2021 [37]	Present Study
**Genotype**	**Nucleotide change**	c.1544T>C + c.3346C>T	c.1841G>A + c. 3239G>C	c.5410_5412delGAG + c.5410_5412delGAG	c.1621G>A + c.1621G>A	c.4819C>T + c.2975_2978delAG	c.1621G>A + c.2382_2383delC	c.4819C>T + c.4819C>T	c.4882C>A + c.5098G>C	c.2521G>A+ c.(897+1_898−1)_(1579+1_1580−1)del
	**Predicted protein change**	p.Ile515Thr + p.Arg1116Ter	p.Gly614Glu + p.Arg1080Pro	p.Glu1804del + p.Glu1804del	p.Gly541Ser + p.Gly541Ser	p.Arg1607Trp + p.Gln994ValfsX6	p.Gly541Ser + p.Leu795SerfsX5	p.Arg1607Trp + p.Arg1607Trp	p.Pro1628Thr + p.Glu1700Gln	p.Glu841Lys
**Phenotype**	**No. patients**	2	1	3	1	1	1	1	4	1
	**Origin**	United States	Brazil	Scotland	Japan	China	China	China	China	Italy
	**Sex**	M (1), F (1)	F	M (1), F (1)	M	M	M	M	M	M
	**Age at onset**	2 years	-	2 years	10 years	13 months	30 months	6 years	8–15 years	2 years
**Afebrile**	**PTA**	Mild HL at low frequencies	Mild HL	Normal/mild HL	Mild HL	Moderate †/mild HL	Moderate HL	Moderate HL †/normal	Normal	Moderate HL at low frequencies
	**SDS (%)**	88–100	-	≤ 80	≤ 80	†/16	93–98 (after CI)	96	88–100	-
	**DWRS**	-	-	-	-	-	-	-	-	100% (quiet) 70% (noise)
	**Ty**	A	-	-	-	A	A	A	A	A
	**SR**	Absent	-	-	-	Absent	Absent	Absent	Absent	Absent
	**OAEs**	Present	Present	Present	Present	Present	Present	Present	Present	Present
	**ABR**	Abnormal	Abnormal	Abnormal	Absent	Absent	Absent	Absent	Normal	Absent
**Febrile**	**Body T (°C)**	38.1/37.8	-	> 38	37.2	36.6 †/36.5	-	36.9	38–40.2	38.7
	**PTA**	Profound/mild HL	Severe	Profound/severe HL	Profound HL	Severe †/moderate HL	-	Mild HL	Mild HL	Moderate HL
	**SDS (%)**	0/0	-	< 100	15	- †/16–20	-	88–80	0–20	-
	**DWRS**	-	-	-	-	-	-	-	-	60% (quiet) 10% (noise)
	**Ty**	A	-	-	-	A	A	A	Ax3/Cx1	A
	**SR**	Absent	-	-	-	Absent	Absent	Absent	Absent	Absent
	**OAEs**	Present	-	Present	Present	Present	Present	Present	Present x 3/ absent x 1	-
	**ABR**	Absent	-	Absent	Absent	Absent	Absent	Absent	Absent	-
	**Therapy**	-	-	Hearing aids	-	Improved with age	Cochlear implantation (CI)	Improved with age	-	Hearing aids

## Data Availability

No new data were created or analyzed in this study. Data sharing is not applicable to this article.

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
