# Peer review of "Temperature-Sensitive Auditory Neuropathy: Report of a Novel Variant of OTOF Gene and Review of Current Literature"

_medicina, 2023, doi:10.3390/medicina59020352_

Round 1

Reviewer 1 Report

Part of the information in the discussion would correspond to the introduction section, some sections could be resume.

There are not electropherograms of the missense pathogenic variant, or the array-CGH 7.4 kb deletion analysis. It would be important to show the Sanger sequencing of the patient, and their parents and the results of the array-CGH analysis. In figure 3, the OTOF protein structure is presented,  it does not describe the section affected by the deletion.

As the pathogenic mutation has been described in other patients without the TS-ANSD, the impact of the pathogenic variants requires discussion in relation to the phenotype and the expected modifications to the protein structure. The authors describe that it was not possible to carry out the array-CGH analysis in the parents, it would be of interest to discuss the segregation analysis carried out in the patient. There are other two variants identified in other genes in the patient that are not further discussed.

Author Response

We thank the Reviewers for the suggestions and the comments. We tried to answer to each point as specifically as possible, and to make the necessary changes in the manuscript text. We hope that this new version is more complete for the readers of Medicina.

REVIEWER 1

Comments and Suggestions for Authors

Part of the information in the discussion would correspond to the introduction section, some sections could be resume.

We thank Reviewer 1 for the comment. To better distinguish the Introduction from the Discussion section, the former being more introductive, the latter being more focused on temperature sensitivity of OTOF and TS-ANSD, we moved line 250-268 of the previous version of the manuscript to the Introduction section (line 56-73).

There are not electropherograms of the missense pathogenic variant, or the array-CGH 7.4 kb deletion analysis. It would be important to show the Sanger sequencing of the patient, and their parents and the results of the array-CGH analysis.

We added figure 3 with the electropherograms and the IGV alignments, showing the variants of the proband in the OTOF gene. The array CGH was performed only for confirm the results of these sequences.

In figure 3, the OTOF protein structure is presented, it does not describe the section affected by the deletion.

We changed the figure and the legend appropriately, adding the protein region affected by the deletion.

As the pathogenic mutation has been described in other patients without the TS-ANSD, the impact of the pathogenic variants requires discussion in relation to the phenotype and the expected modifications to the protein structure.

We tried to address this issue in the paragraph comprised between line 299 and line 323.

The authors describe that it was not possible to carry out the array-CGH analysis in the parents, it would be of interest to discuss the segregation analysis carried out in the patient.

The parents' DNA was not available for segregation analysis.

There are other two variants identified in other genes in the patient that are not further discussed.

We tried to address this issue in the paragraph comprised between line 299 and line 323.

Moreover, English was revised and some minor linguistic changes were made.

Reviewer 2 Report

Dear Ladies and Gentlemen, Dear-Journal-Team,

the manuscript "Temperature-sensitive Auditory Neuropathy: review of current literature and report of a novel variant of OTOF gene" is well written. Tables and Figures are sufficient.

1. It would be interesting to discuss that the auditory brain stem response recording and the small reflexes are dependent on synchrony of the same nerve and nerve groups and distinct amplitude levels.

2. Please explain every abbreviation, when first used: Cy5-dCTP (2.4.3. Array Comparative Genomic Hybridization), HGMD (Human Gene Mutation Database, 2.5. Genetic results), GJB2 (Gap junction protein beta 2/connexin 26, autosomal recessive, 2.5. Genetic results), TECTA (alpha-tectorin protein, 2.5. Genetic results), ValfsX6 and SersX6 (Table 1).

3. Please change in the discussion, line 230 to 'pure tone audiometry'.

4. Please check Reference 33, Rodriguez-Ballesteros et al. for accuracy of the author names. Check Reference 40 Wang et al. for style unformity of the DOI reference.

Sincerely,

Author Response

We thank the Reviewers for the suggestions and the comments. We tried to answer to each point as specifically as possible, and to make the necessary changes in the manuscript text. We hope that this new version is more complete for the readers of Medicina.

REVIEWER 2

Comments and Suggestions for Authors

Dear Ladies and Gentlemen, Dear-Journal-Team,

the manuscript "Temperature-sensitive Auditory Neuropathy: review of current literature and report of a novel variant of OTOF gene" is well written. Tables and Figures are sufficient.

  1. It would be interesting to discuss that the auditory brain stem response recording and the small reflexes are dependent on synchrony of the same nerve and nerve groups and distinct amplitude levels.

We thank Reviewer 2 for the comments. We reformulated the sentence regarding ABRs and Stapedial Reflexes, which both were absent even at maximum stimulation levels. This is in line with what is commonly described in patients affected by OTOF-related auditory neuropathy.

  1. Please explain every abbreviation, when first used: Cy5-dCTP (2.4.3. Array Comparative Genomic Hybridization), HGMD (Human Gene Mutation Database, 2.5. Genetic results), GJB2 (Gap junction protein beta 2/connexin 26, autosomal recessive, 2.5. Genetic results), TECTA (alpha-tectorin protein, 2.5. Genetic results), ValfsX6 and SersX6 (Table 1).

All the abbreviations have been explained. However, we could not express differently p.Gln994ValfsX6 and p.Leu795SerfsX5 (Table 1) as they are genetic abbreviations (for example “p.Arg83SerfsX15” means that arginine is the first amino acid changed, it is in position 83, it makes serine instead, the length of the shift frame is 15, including the stop codon (X), as taken from the website https://atlasgeneticsoncology.org/teaching/30067/nomenclature-for-the-description-of-mutations-and-other-sequence-variations accessed on 24th, January 2023).

  1. Please change in the discussion, line 230 to 'pure tone audiometry'.

It is not clear what reviewer 2 means with this comment, as in line 230 is not reported any term that could be corrected to “pure tone audiometry”.

  1. Please check Reference 33, Rodriguez-Ballesteros et al. for accuracy of the author names. Check Reference 40 Wang et al. for style unformity of the DOI reference.

Both references were checked and corrected.

Sincerely,

Reviewer 3 Report

This paper reports two pathogenic variants that have never been associated with temperature-sensitive auditory neuropathy, reviewing previous case reports of temperature-sensitive auditory neuropathy and studies on the temperature sensitivity of otoferlin. The paper is well organized.

Some suggestions are as follows:

1. line 35-36: It seems inappropriate to refer to this patient as an "asymptomatic proband".

2. Figure 1 and 2: The meaning of the different colored lines in the figures should be marked.

3. Table 1: The age of onset of the case reported in this paper in the table appears to be inconsistent with the age of onset described in other parts of the paper.

4. whether "N° patients" in the phenotype section of Table 1 is a clerical error.

Author Response

We thank the Reviewers for the suggestions and the comments. We tried to answer to each point as specifically as possible, and to make the necessary changes in the manuscript text. We hope that this new version is more complete for the readers of Medicina.

REVIEWER 3

Comments and Suggestions for Authors

This paper reports two pathogenic variants that have never been associated with temperature-sensitive auditory neuropathy, reviewing previous case reports of temperature-sensitive auditory neuropathy and studies on the temperature sensitivity of otoferlin. The paper is well organized.

Some suggestions are as follows:

  1. line 35-36: It seems inappropriate to refer to this patient as an "asymptomatic proband".

We thank Reviewer 3 for the comment. The sentence was ambiguous, as by “The asymptomatic proband’s parents” we meant that the parents of the proband were asymptomatic, not the proband himself. The sentence was reorganized to avoid ambiguity.

  1. Figure 1 and 2: The meaning of the different colored lines in the figures should be marked.

A legend was added to the figures representing the two audiograms (Fig. 1 and 2), and to the figures’ caption.

  1. Table 1: The age of onset of the case reported in this paper in the table appears to be inconsistent with the age of onset described in other parts of the paper.

We thank the reviewer for highlighting this mismatch. The table was therefore corrected, as the symptoms emerged in the patient for the first time at the age of 2, as reported by the parents, while the episode we were able to record happened when the patient was 8.

  1. whether "N° patients" in the phenotype section of Table 1 is a clerical error.

We changed N° into No.

Round 2

Reviewer 3 Report

The authors answered all the questions successfully.

Author Response

We thank Reviewer 3 for the time and effort in revising our manuscript.